# Comparison of Transcriptome between Tolerant and Susceptible Rice Cultivar Reveals Positive and Negative Regulators of Response to *Rhizoctonia solani* in Rice

**DOI:** 10.3390/ijms241814310

**Published:** 2023-09-20

**Authors:** Xiurong Yang, Shuangyong Yan, Yuejiao Li, Guangsheng Li, Shuqin Sun, Junling Li, Zhongqiu Cui, Jianfei Huo, Yue Sun, Xiaojing Wang, Fangzhou Liu

**Affiliations:** 1Institute of Plant Protection, Tianjin Academy of Agricultural Sciences, Tianjin 300381, China; 2Institute of Crop Research, Tianjin Academy of Agricultural Sciences, Tianjin 300381, China

**Keywords:** *Oryza sativa*, rice sheath blight, transcriptome, positive regulator, negative regulator

## Abstract

Rice (*Oryza sativa* L.) is one of the world’s most crucial food crops, as it currently supports more than half of the world’s population. However, the presence of sheath blight (SB) caused by *Rhizoctonia solani* has become a significant issue for rice agriculture. This disease is responsible for causing severe yield losses each year and is a threat to global food security. The breeding of SB-resistant rice varieties requires a thorough understanding of the molecular mechanisms involved and the exploration of immune genes in rice. To this end, we conducted a screening of rice cultivars for resistance to SB and compared the transcriptome based on RNA-seq between the most tolerant and susceptible cultivars. Our study revealed significant transcriptomic differences between the tolerant cultivar ZhengDao 22 (ZD) and the most susceptible cultivar XinZhi No.1 (XZ) in response to *R. solani* invasion. Specifically, the tolerant cultivar showed 7066 differentially expressed genes (DEGs), while the susceptible cultivar showed only 60 DEGs. In further analysis, we observed clear differences in gene category between up- and down-regulated expression of genes (uDEGs and dDEGs) based on Gene Ontology (GO) classes in response to infection in the tolerant cultivar ZD, and then identified uDEGs related to cell surface pattern recognition receptors, the Ca^2+^ ion signaling pathway, and the Mitogen-Activated Protein Kinase (MAPK) cascade that play a positive role against *R. solani.* In addition, DEGs of the jasmonic acid and ethylene signaling pathways were mainly positively regulated, whereas DEGs of the auxin signaling pathway were mainly negatively regulated. Transcription factors were involved in the immune response as either positive or negative regulators of the response to this pathogen. Furthermore, our results showed that chloroplasts play a crucial role and that reduced photosynthetic capacity is a critical feature of this response. The results of this research have important implications for better characterization of the molecular mechanism of SB resistance and for the development of resistant cultivars through molecular breeding methods.

## 1. Introduction

Rice (*Oryza sativa* L.) is one of the world’s most crucial food sources, providing sustenance for more than half of the global populace [1]. Nonetheless, the crop faces several challenges, including the overapplication of fertilizers and pesticides, climate change, limited availability of resistant cultivars, and the prevalence of devastating diseases like sheath blight (SB) caused by *Rhizoctonia solani* [2,3]. This soil-borne necrotrophic fungus boasts a vast array of hosts, capable of affecting 32 plant families and 188 genera [4]. The pathogen employs hyphal aggregates called infection cushions during infection initiation [5]. By secreting toxins and enzymes, *R. solani* triggers cell death before colonizing host tissues [6]. To survive winter seasons, this pathogen forms sclerotia, which serve as a primary source of infection in the following season [7]. Despite significant efforts directed towards understanding the resistance mechanisms of diverse rice varieties against *R. solani*, there exist low-tolerance rice cultivars, impeding the growth and expansion of rice farming on a global scale [8]. Thanks to advancements in molecular biology and the widened utilization of omics technologies in examining interactions between plant pathogens, the discovery of rice SB resistance genes and their corresponding interaction mechanism with *R. solani* has become achievable [8].

The disease resistance proteins OsRSR1 and OsRLCK5 interact with OsSHM1 (a serine hydroxy methyltransferase) and OsGRX20 (a glutaredoxin), respectively, to regulate reactive oxygen species (ROS) levels and enhance resistance to *R. solani* [9]. Moreover, another positive regulator of SB resistance, WRKY30, positively controls jasmonic acid (JA) biosynthesis by activating related genes and stimulating endogenous JA accumulation [10]. Two other important regulatory components of rice pathogen responses, namely *WRKY4* and *WRKY80*, also play essential roles in defense signaling [11]. Additionally, lignification reinforces plant cells against microbial invasion, leading Tonnessen et al. [12] to suggest that up-regulating the lignin-synthesizing gene *OsPAL4* could increase resistance to SB in rice. Finally, it was observed that *R. solani* infections trigger the production of OsPGIP1, a polygalacturonase-inhibiting protein, which confers increased anti-fungal properties and promotes SB resistance when overexpressed in transgenic rice plants [13].

With the advent of advanced technologies for studying host–pathogen interactions, RNA sequencing (RNA-Seq) technology has emerged as a highly effective tool due to its capability for large-scale analysis at reduced costs in genomic analysis and gene function research [14]. The results revealed that regulation of the JA signaling and phenylpropanoid pathways, photosynthesis, and photorespiration may contribute to rice resistance to SB when the leaf transcriptome changes in response to *R. solani* infection between TeQing (a moderately resistant cultivar) and Lemont (a susceptible cultivar) [14]. Further investigations using transcriptome analysis showed that the response of the resistant cultivar Shennong 9819 to pathogen stress was faster than that of the susceptible cultivar, and that a pathogen-induced defense system involved in defense-related pathway genes, pathogenesis-related (PR) genes, key transcription factors, and phenylalanine ammonia-lyase genes was activated in resistant rice cultivars in the early stages of *R. solani* infection [15].

In order to breed rice varieties with stable resistance to SB, it is necessary to gain insight into the molecular mechanisms based on a broad exploration of innate defense genes in rice [3]. Therefore, we conducted a screening of rice cultivars for resistance to SB by manual inoculation method to discover disease-resistant rice cultivars, and compared the transcriptome based on RNA-seq between the most tolerant and susceptible cultivar response to *R solani* to discover resistance genes and dissect the molecular mechanism. We found significant differences in the transcriptomic response to *R. solani* infection between ZD and XZ. Furthermore, we identified candidate genes for the activation and inhibition of regulators in different biological processes and pathways. This may be useful in the future for the molecular breeding of rice cultivars with high SB resistance.

## 2. Results

### 2.1. Screening of Rice Varieties for Sheath Blight Resistance

The main symptoms observed in the rice leaf sheath after 4 weeks of inoculation are shown in Figure 1. Among 22 rice cultivars screened for SB, only three cultivars (ZD, XinKeDao 22, and XinLiang 320) were found to be tolerant to *R. solani* with a range of less than 30% (Figure 1 and Table 1). There were five cultivars (XinDao575, YueNongDao No.1, ShengDao735, YiDao 735, and ZhengDao 25) moderately susceptible to *R. solani*, the percentage of RLH in the range of 31–45%. XZ, YiDao 178, and Xu72985 showed high susceptibility with the range of 16–30%. Eleven other cultivars showed susceptibility to *R. solani* with the lesion height of the leaf sheath covering more than 46% of the plant height.

After the primary screening of rice cultivars, the most susceptible cultivar, XZ, and the most tolerant cultivar, ZD, were selected for further analysis. By direct observation, both XZ and ZD showed typical SB symptoms on the leaf sheath after inoculation for 72 h (Figure 2A). The expanded spot area of susceptible XZ was significantly larger than that of tolerant ZD (Figure 2B).

### 2.2. RNA-Seq Data and Alignment Analysis 

The high accuracy of the RNA-seq analysis was demonstrated by the average of 93.87% of the total reads passing ≥30 Phred scores, and the average base composition and quality of the total samples. A reference-based pairwise alignment with the rice IRGSP1.0 reference genome was performed on the pre-processed and rRNA-removed reads. Appendix A shows the overall alignment summary and the fastq file summary for the samples. It showed that approximately 75.5% of clean reads successfully aligned to the rice reference genome. This suggests that the data are suitable for further analysis.

### 2.3. Differential Expression Gene Analysis of Tolerant and Susceptible Cultivar Response to R. solani

The RNA-seq profile showed 4141 up-regulated genes and 2925 down-regulated genes in the infected tolerant ZD compared to the uninfected control, whereas in susceptible XZ, the number of up- and down-regulated genes was only 52 and 8, respectively, in sharp contrast to the tolerant ZD (Table 2, Figure 3, Appendix A). Thus, there was a large difference in the response to *R. solani* between the tolerant rice variety ZD and the susceptible variety XZ. This result indicated that the immune defense response was successfully initiated in the resistant variety ZD in response to *R. solani* invasion, resulting in differential expression of many relevant genes. In contrast, the resistance responses could not be initiated in the susceptible variety XZ.

### 2.4. GO Enrichment Analysis of DEG Response to R. solani in Tolerant Cultivar ZD

Therefore, we further analyzed the DEGs of the disease-resistant cultivar ZD in response to the fungus invasion. These 4141 up-regulated DEGs (uDEGs) and 2925 down-regulated DEGs (dDEGs) in response to *R. solani* in the tolerant cultivar ZD were subjected to Gene Ontology (GO) analysis. There were significant differences in gene category between uDEGs and dDEGs after GO enrichment analysis (Appendix A). Enrichment analysis of DEGs was performed based on the Biological Process (BP) of GO, which showed that these uDEGs hit 367 BPs level 2 classes, while dDEGs hit 134 BPs level 2 classes based on enrichment FDR cutoffs 0.05 (Appendix A). Surprisingly, there was no overlap between uDEGs and dDEGs in BP classes. The top ten BP classes on uDEGs were “Small Molecule Metabolic Process”, “Phosphorylation, Organic Acid Metabolic Process”, “Carboxylic Acid Metabolic Process”, “Oxo Acid Metabolic Process”, “Protein Phosphorylation”, “Monocarboxylic Acid Metabolic Process”, “Carbohydrate Metabolic Process”, “Organic Acid Biosynthetic Process”, and “Response to Biotic Stimulus”, suggesting that uDEGs involved in these biological processes play a positive role. The top ten BP classes for dDEGs were “Photosynthesis”, “Response to light stimulus”, “Response to radiation”, “Plastid organisation”, “Photosynthesis, light response”, “Chloroplast organisation”, “Photosynthesis, light harvesting in photosystem I”, “Photosynthesis, light harvesting”, “Protein-chromophore linkage”, and “Response to abiotic stimulus”. Among them, the enrichment FDR value of photosynthesis was the lowest, suggesting that dDEGs are involved in the biological process of photosynthesis and play a negative role (Figure 4).

Enrichment analysis of the DEGs was performed based on the cellular component (CC) of GO, where these uDEGs encountered 63 CC level 2 classes, whereas dDEGs encountered 37 CC classes (Appendix A). Similar to the BP-based enrichment analysis, there were only two class overlaps between the uDEGs and dDEGs in the CC classes, i.e., “organelle membrane” and “organelle subcompartment”. The top ten CC classes for uDEGs were “endomembrane system”, “golgi apparatus”, “anchored component of membrane”, “organelle membrane”, “anchored component of plasma membrane”, “respirasome”, “intrinsic component of plasma membrane”, “golgi apparatus subcompartment”, “mitochondrial envelope”, and “mitochondrial inner membrane”, suggesting that uDEGs located in the endomembrane system play a positive role, while the top ten CC classes for dDEGs were “plastid”, “chloroplast”, “thylakoid”, “chloroplast thylakoid”, “plastid thylakoid”, “plastid envelope”, “thylakoid membrane”, “chloroplast thylakoid membrane”, “plastid thylakoid membrane”, and “photosynthetic membrane”. This indicated that dDEGs located in the chloroplast play a negative role (Figure 4).

The enrichment analysis of these DEGs was performed based on the molecular function (MF) of GO, where these uDEGs hit 150 MF classes, while dDEGs hit only 8 MF classes. There was a very clear difference between the uDEGs and dDEGs based on MF classes. The top ten MF classes for uDEGs were “kinase activity”, “phosphotransferase activity, alcohol group as acceptor”, “transferase activity, transferring phosphorus-containing groups”, “protein kinase activity”, “protein serine/threonine kinase activity”, “oxidoreductase activity”, “calcium ion binding”, “carbohydrate binding “, “transporter activity”, and “glycosyltransferase activity”, while the eight MF classes of for dDEGs were “oxidoreductase activity”, “tetrapyrrole binding”, “iron ion binding”, “protein heterodimerization activity”, “quercetin 3-O-glucosyltransferase activity”, “quercetin 7-O-glucosyltransferase activity”, “chlorophyll binding”, and “protein domain specific binding” (Appendix A).

The results showed that the tolerant ZD cultivar responded to *R. solan* by activating or inhibiting genes involved in a number of different biological processes occurring in different cellular components. In addition, the activated genes exhibited a wide range of molecular functions, characterized by considerable diversity.

### 2.5. KEGG Enrichment Analysis of DEG Response to R. solani in Tolerant Cultivar ZD

Enrichment analysis of DEGs responsive to *R. solani* in the tolerant cultivar ZD was performed based on the Kyoto Encyclopedia of Genes and Genomes (KEGG) databases and revealed that these uDEGs and dDEGs were significantly enriched in 51 and 5 KEGG pathways, respectively (Figure 5, Appendix A). Among them, “metabolic pathways”, “biosynthesis of secondary metabolites”, “carbon metabolism”, “biosynthesis of amino acids”, and “plant-pathogen interaction” were the five pathways most enriched in uDEGs, while “signal transduction of plant hormones”, “photosynthesis”, “glyoxylate and dicarboxylate metabolism”, “carbon fixation in photosynthetic organisms” and “carotenoid biosynthesis” were the KEGG pathways significantly enriched with dDEGs (Figure 5). Three KEGG pathways, “plant hormone signal transduction”, “glyoxylate and dicarboxylate metabolism”, and “carbon fixation in photosynthetic organisms”, overlapped with uDEGs and dDEGs in the KEGG pathway classes. This implied that DEGs may play a positive or negative role in the response to *R. solani* in the tolerant cultivar ZD in these pathways.

### 2.6. Analysis of DEGs of Transcription Factors Response to R. solani in Tolerant Cultivar ZD

It is well known that transcription factors (TFs) play an important role in plant disease resistance response; therefore, we further analyzed the role of TFs against *R. solani*. In the present study, a total of 2002 TFs were detected in the rice transcriptome of ZD and XZ, which were classified into 79 families (Appendix A). The top 20 TF families are exhibited in Figure 6A. Among them, AP2-EREBP (142), bHLH (117), MYB (105), NAC (92), and C2H2 (84) are the top five TF families, accounting for more than 27% of the total number of TFs (Figure 6A).

In the susceptible cultivar XZ, only three DEGs of TFs (DEFs) were identified: the bZIP family TF Os08g0487100, the NAC family TF Os01g0925400, and the SRS family TF Os01g0954500. However, in the tolerant cultivar ZD responding to *R. solani*, a total of 470 DEFs from 58 TF families were detected, of which 260 were up-regulated and 210 were down-regulated (Appendix A). KEGG enrichment analysis of the up-regulated DEFs revealed enrichment in plant hormone signaling (23), MAPK signaling (9), and plant–pathogen interaction (4), whereas the down-regulated DEFs were enriched only in plant hormone signal transduction (Appendix A). The 14 major TF families of DEFs are shown in Figure 6B. Among them, AP2-EREBPs (48; 42 up-regulated and 6 down-regulated), WRKYs (35; 32 up-regulated and 6 down-regulated), MYBs (30; 26 up-regulated and 4 down-regulated), bHLHs (30; 18 up-regulated and 12 down-regulated), and NACs (29; 24 up-regulated and 5 down-regulated) are the top five TF families, accounting for more than 36% of the total number of DEFs (Figure 6B, Appendix A).

### 2.7. Validation of Differentially Expressed Genes through qRT-PCR

To validate the RNA-seq data, qRT-PCR was performed with 15 selected uDEGs and dDEGs of the tolerant cultivar ZD. In the qRT-PCR analysis, 15 genes showed concordant results with the RNA-seq analysis. The expression pattern was similar in both analyses, but the fold change in qRT-PCR was lower compared with RNA-seq data (Figure 7).

## 3. Discussion

A pathogen-associated molecular pattern (PAMP) recognized by a pattern recognition receptor (PRR) localized at the cell surface triggers a host-induced defense response that confers basal resistance to resistant hosts. Upon pathogen PAMP recognition, resistant hosts directly trigger PTI, such as activation of the Ca^2+^ pathway and activation of the mitogen-activated protein kinase (MAPK) cascade. Then, the downstream phytohormone signaling pathway, such as salicylic acid (SA) or JA/ET, is activated, leading to the activation of disease resistance responses [16]. In this study, differential transcriptome analysis was performed on inoculated and noninoculated samples of resistant and susceptible rice cultivars. The results showed that the resistant cultivar ZD, with a very high number of DEGs, was able to induce PTI and then activate an immune response against *R. solani* invasion, whereas the susceptible cultivar XZ, with a very low number of DEGs, was unable to induce an immune response. Further analysis revealed that uDEGs of many PRRs and several immune-related pathways positively regulated disease resistance in the resistant rice cultivar ZD, while DEGs involved in phytohormone signaling acted as positive or negative regulators and dDEGs involved in photosynthesis and chloroplast negatively regulated disease resistance. TFs are also important for the response to fungal invasion. We have summarized the information about these possible positive or negative regulators in Appendix A.

### 3.1. Activation of Cell Surface Pattern-Recognition Receptors 

Plants activate PTI through PRRs by recognizing conserved PAMPs. PRRs are typically either receptor-like kinases (RLKs) or receptor-like proteins lacking a protein kinase domain [17]. RLKs are critical components of signaling cascades that enable plants to sense their environment and initiate appropriate adaptive responses, and have been implicated in many aspects of plant development, growth, and response to biotic and abiotic stresses [18]. RLKs are classified into subfamilies, including lysine motif kinases (LysM), receptor-like cytoplasmic kinases (RLCK), wall-associated kinases (WAKs), S-domain subfamily of receptor-like kinases (SDRLKs), and others [17,19].

CHITIN ELICITOR RECEPTOR KINASE 1 (CERK1) is a LysM kinase known to be a protein component of PRRs in plants (Shimizu et al., 2010) [19]. OsCERK1 interacts with CEBiP, a receptor-like protein, to recognize chitin oligomers and peptidoglycan, which are PAMPs of fungi and bacteria, and to mediate signaling pathways to participate in innate PTI when infected with *Magnaporthe oryzae* in rice [20,21].

RLCKs contribute to cytoplasmic phosphorylation signaling pathways in PTI [22]. Members belonging to the VII of the RLK family are crucial mediators in the transmission of signals originating from membrane PRRs to downstream effector molecules in plant immune response [23]. Several members of the RLCK family have been shown to play positive regulatory roles in rice disease resistance. Members of the RLCK subfamily, the *BSR1*, *OsRLCK57*, and *OsRLCK107* genes, contribute to chitin-triggered defense responses in rice; in particular, *OsRLCK5* positively regulates resistance to *R. solani* [9,21,22,24].

WAKs are characterized by an extracellular domain consisting of one or more repeats of the epidermal growth factor domain, which is known to dimerize and bind small peptides upon calcium binding [25]. WAK has emerged as a positive regulator of fungal disease resistance in several plant species, but the pathways involved in this regulation are unknown [25]. A total of 125 genes encoding OsWAK proteins have been identified in rice, but the functions of most of these genes are unknown [26]. *OsWAK14*, *OsWAK91*, and *OsWAK92* are positive regulators of quantitative resistance, and *OsWAK112d* is a negative regulator of rice blast resistance [25].

The SDRLK subfamily is the second-largest subfamily of RLKs and shares similarities with the extracellular domain of the S locus receptor kinase protein [18,27]. The rice SDRLK subfamily comprises 144 members, and most of their functions are not yet known. However, these genes are increasingly found to be expressed in different tissues and in response to pathogen resistance in rice [28]. *PID2* and *SDS2*, two genes of the rice SDRLK subfamily, contribute to the positive regulation of pathogen resistance and immunity [28,29]. Naithani et al. analyzed the expression datasets of 39 genes of the SDRLK family from *R. solani* strain LR172-infected resistant and susceptible rice. A total of 14 genes of the rice SDRLK family were found to show differential expression in response to *R. solani* [27].

In our enrichment analysis of uDEGs based on MF, we found that uDEGs are enriched in the GO term “kinase activity,” which is the most enriched uDEG in the MF term (Appendix A). Plant RLKs account for 60% of kinases in the Arabidopsis genome, and kinase activity plays a very important role in the plant immune system [17]. uDEGs with kinase activity belong to the subfamily of LysMs (3), RLCKs (63), SDRLKs (23), and WAKs (21), so it is very reasonable to assume that these uDEGs play a positive regulator of active downstream metabolic pathways when the torrent cultivar is infected with *R. solaniis* (Appendix A).

### 3.2. Activation of Ca^2+^ Signaling Pathway in Response to R. solani

During biotic or abiotic stress, the Ca^2+^ concentration in the cytosol of the cell increases [30,31]. Maintenance of cytosolic Ca^2+^ concentration is mediated by different classes of calcium entry and exit proteins [32]. Therefore, Ca^2+^ accumulated in the cytosol during a plant–microbe interaction can generate distinct and precise calcium signatures that are recognized by different calcium-sensing proteins and directly control cellular redox homeostasis to regulate calcium ion-dependent gene expression [33]. In plants, Ca^2+^ is transported by several channel proteins, including annexins, bipolar channels, mechanosensitive channels, and ionotropic glutamate receptors [33].

Several classes of Ca^2+^-sensing proteins, containing EF-hand motifs that can sense Ca^2+^ ions, are responsible for detecting and interpreting calcium signals, which are then transduced through specific downstream signaling pathways [33]. In plants, there are four main categories of calcium sensors: calcium calmodulins (CaMs), CaM-like proteins (CMLs), calcineurin B-like proteins (CBLs), and calcium-dependent protein kinases (CDPKs) [34].

The globular domains of CaM proteins consist of two EF-hands separated by a flexible helix and are highly conserved throughout the plant kingdom [35]. These proteins can bind up to four Ca^2+^ ions, and downstream signaling is mediated by various calmodulin-sensing proteins. CaM can trigger various physiological responses both alone and in concert with other calmodulin-sensing proteins [33]. Recently, research has shown that the CaM-binding CBP60g family is activated when exposed to fungal and bacterial pathogens and plays a central role in immunity in rice and Arabidopsis [36].

CML is a distinct class of Ca^2+^-sensing proteins with a 148-amino-acid sequence difference from CaM and minimal similarity with CaM [37]. CML are unique sensor relay proteins in plants and consist of two to six Ca^2+^-binding EF-hand motifs with a broad range of functions from development to stress response in plants [38]. A member of the CML family, CML8, showed resistance to Pseudomonas syringae in Arabidopsis via the SA-mediated PR1 activation pathway in Arabidopsis [39].

CBL proteins activate their downstream target proteins, CBL-interacting protein kinases (CIPKs), in a variety of biological processes [40,41]. CBLs and CIPKs are two relatively new but important classes of plant calcium sensors [37]. A total of 12 CBL genes have been identified in maize, and most of them are involved in abiotic stress tolerance. While the majority of CBL and CIPK functions have been associated with drought, salt, and other abiotic stress tolerance, their emerging role in biotic stress response has become increasingly clear [33]. Studies have shown that OsCIPK14 and OsCIPK15 in rice are up-regulated in response to PAMP treatment and confer resistance through the activation of ROS-mediated HR and cell death [42].

CDPKs constitute the most abundant class of calcium sensors in plants [33]. The intracellular Ca^2+^ level is a major determinant of the activity of CDPKs. At low levels of intracellular calcium, the auto-inhibitory domain binds to the kinase domain of CDPKs, thereby limiting the phosphorylating activity of CDPKs on its target protein. When cell Ca^2+^ levels rise, EF’s hands start binding these calcium ions, which frees its kinase domain for phosphorylating its target. Paralogous CDPKs (CDPK3 and CDPK4) from barley (*Hordeum vulgare*) are capable of inhibiting the penetration of *powdery mildew* when expressed in tobacco [43]. In most cases, CDPKs trigger HR-mediated cell death by inducing oxidase-dependent ROS production in infected cells, providing an interesting mechanism of action of the pathogens [44].

In our enrichment analysis of uDEGs based on MF, we found that uDEGs are also enriched in “calcium ion binding” and “transporter activity”, which are the top 10 enriched uDEGs based on enrichment FDR (Appendix A). Many Ca2+ signaling pathway genes, including 4 of channel proteins, 4 of CAMs, 1 of CBL, 5 of CIPKs, 12 of CDPKs, and 13 of CMLs, were up-regulated, suggesting that the Ca2+ signaling pathway and these uDEGs are very important to regulate resistance to this pathogen infection (Appendix A).

### 3.3. Activation of MAPK Cascade

The MAPK cascade, which consists of MAPK kinase kinases (MAPKKs), MAPK kinases (MAPKKs), and MAPKs, plays an important role in the regulation of plant defense responses through sequential phosphorylation [45,46]. After phosphorylation, MAPKs translocate to the nucleus where they can target other kinases, proteins, or transcription factors to initiate a downstream defense response [47]. Alternatively, MAPKs are involved in phytohormone accumulation and signal transduction, including but not limited to abscisic acid (ABA), SA, JA, ET, brassinosteroids (BR), and cytokinin (CK), as reported by Chen et al. [48] The rice genome contains 74 MAPKKK, 8 MAPKK, and 17 MAPK genes. Many of the genes in the MAPK cascade in plants regulate the defense response to pathogen invasion, either in a positive or negative manner [49].

MAPK cascade genes play an important role in the defense response of rice against pathogen infections. For example, the OsMPKKKε-OsMPKK4/5-OsMPK3/6 cascade, the OsMPKK4-OsMPK3/6 cascade, and the OsMAPKKK11/18/24-OsMAPKK4/5-OsMAPK3/6 cascade have been shown to regulate chitin signaling and increase resistance to rice blast [50]. OsMKK10-2-mediated activation of OsMPK6 induces WRKY45 expression and improves resistance to *M. oryzae* in rice [51]. In addition, OsEDR1 (OsMPKKK1) is known to positively regulate resistance to *M. oryzae*, but reduces it to *Xoo* by activating ethylene synthesis [52], and OsMPK4 and OsMPK17-1 have also been shown to contribute to resistance to *Xoo* infection [53], whereas OsMPK15 negatively regulates resistance to both the fungus *M. oryzae* and the bacterium *Xoo* [54].

In our current analysis, we found that there were five MAPKKKs, three MAPKKs, and five MAPKKs in uDEGs with kinase activity responses to *R. solaniis* in the tolerant cultivar ZD, indicating that the MAPK cascade works against *R. solani* infection (Appendix A). The MAPK cascade plays an important role in rice blast resistance, but the role of MAPK cascade in defense against SB is still relatively little reported. The findings of this study provide an important research basis to further elucidate the role of MAPK cascade in resistance to SB in rice.

### 3.4. Positive and Negative Regulators in Plant Hormone Signal Transduction

Plant hormones play a central role in triggering immune responses by regulating PTI signaling pathways, with SA, JA, and ET being the most important signals [16]. Resistant cultivars respond to pathogen invasion by activating or repressing key signaling system genes. In our study, both uDEGs and dDEGs were enriched in plant hormone signaling, so these genes could be positive or negative regulators in resistant responses (Appendix A).

JA is a lipid-derived phytohormone and is considered an essential stress signaling molecule that begins to accumulate massively and rapidly in plant tissues when they are attacked by microbial pathogens or insects [55]. JA has the potential to induce the expression of several defense-related proteins and stimulate the production of certain volatiles, alkaloids, and the formation of defense structures, thus playing an important role in conferring stress and disease resistance functions to plants [56]. The synthesis of JAs occurs when α-linolenic acid (α-LeA) from chloroplast membranes undergoes oxidative reactions along different branches of the lipoxygenase pathway [56]. In our study, uDGEs enriched in the KEGG pathway α-linolenic acid metabolism and oxidative phosphorylation processes, which are two of the top ten enriched KEGG pathways, suggest that JA may be a component of the response to *R. solani* infection (Appendix A).

JAZ, which represses regulators of JA signaling, interacts with the transcriptional activator *MYC2* and simultaneously represses negative regulators of JA signaling, known as bHLH transcriptional repressors III [56]. In our analysis, *OsMYC2* expression was up-regulated, but several JAZs were also up-regulated, suggesting that these JAZs may act as negative regulators of transcriptional repressors of JA signaling (Appendix A).

ET plays a vital regulatory function in numerous physiological processes of plants by the ET response pathway [57]. CTR1 is a serine/threonine protein kinase, and a negative regulator of the downstream ET response pathway [16,57]. Binding to ET, which leads to the subsequent inactivation of CTR1, initiates the transmission of a signal to the transcription factor EIN3, then activates ET response factors, such as ERF1 and ERF83, leading to the positive regulation of disease resistance to *M. oryzae* in rice [58]. In our results, the negative regulator CTR1 was down-regulated; EIN3 and two ET response factors, ERF87 and ERF87, were up-regulated. In addition, ERF1 can activate the downstream ET response by interacting with the GCC box in the promoter of target genes *EIL1* and *EIL2* [59]. The two target genes were also up-regulated in our study (Appendix A).

Rhizosphere microorganisms can induce systemic resistance, which can significantly enhance plant resistance to a broad spectrum of pathogens through the joint mediation of JA and ET [16]. In the JA/ET signal transduction pathway, ERF1 and MYC2 act as signal integrators and are responsible for the activation of defense-related genes, which encode antimicrobial peptides involved in the JA/ET response [60]. In our study, defense-related genes such as *PR1B*, *PR1-12*, *PR1A*, and *OsPR1* were also up-regulated. Thus, we suggest that the JA/ET signaling pathway is essential in the SB resistance response (Appendix A).

Auxin plays an important role in plant–microbe interactions, particularly those between plant hosts and disease-causing pathogenic microorganisms, and is one of the major plant hormones that significantly influences many aspects of plant growth and development [61]. It has been shown that indole-3-acetic acid (IAA), the most studied form of auxin, can act both as a plant hormone modulating host signaling and physiology to increase host susceptibility and as a microbial signal directly acting on the pathogen to enhance virulence [61]. In our present investigation, 27 dDEGs were enriched in the plant hormone signal transduction pathway and 14 dDEGs were involved in the auxin signaling pathway response to *R. solani* infection, indicating that these dDEGs may act as negative regulators, in agreement with previous research [61].

### 3.5. Activation and Inhibition of Transcription Factors

Various plant-specific TFs play a critical role in coordinating plant immune gene expression. Generally, these TFs act downstream of MAPK cascades or Ca^2+^ signals triggered by various activation mechanisms in response to pathogenic infection [62]. Many TFs involved in rice defense mechanisms have been identified to regulate the expression of defense genes [62,63], such as *Pibp1* [64], *SPL6* [65], *bHLH84* [66], etc.

Based on our findings, 470 transcription factors are potentially involved in *R. solani* infection in the resistant variety ZD, including 260 uDEGs and 210 dDEGs. In contrast, only three transcription factors were differentially expressed in the susceptible varieties after inoculation, indicating that transcription factors are actually related to the response to SB in rice.

In this study, we identified 60 TF gene families that were differentially expressed in the resistant varieties. Most of the TF families contained uDEGs and dDEGs, which may play a positive or negative role. Members of the TF families AP2-EREBP, WRKY, and MYB were mainly positive regulators. Eleven DEGs of the Tify TF family were all up-regulated, whereas eleven DEGs of the mTERF TF family were all down-regulated (Appendix A).

The APETALA2/ethylene-responsive element binding protein (AP2/EREBP) superfamily is one of the largest and most specific TF families in plants, which has been implicated in a variety of responses to environmental stresses such as cold, heat, drought, high salt, and pathogen infection, including direct stress response and regulation of downstream gene expression. AP2/EREBP TFs are divided into four subfamilies: the ethylene-responsive factor (ERF) subfamily, the dehydration-responsive element binding protein (DREB) subfamily, the APETALA2 (AP2) subfamily, and the ABI3/VP1 (RAV) related subfamily [67]. In our study, 31 of the 42 uDEGs of AP2/EREBP TFs belong to the ERF subfamily. The members of the ERF subfamily bind directly to GCC boxes (AGCCGCC) and regulate the expression of PR genes. ERFs are also involved in plant hormone signaling pathways, such as the ET and JA pathways, which are important for plant stress responses [67]. These uDEGs, especially ERFs, have been shown to play a positive role in the response to SB disease in resistant rice cultivars.

WRKYs encode proteins that bind to the cis-acting element W-box and have been found to be involved in the regulation of defense genes in various plants [68]. Numerous reports have shown that WRKY genes are involved in defense responses and hormone regulation in rice, such as *OsWRKY13*, *OsWRKY71*, *OsWRKY13*, *OsWRKY45*, *OsWRKY51*, and *OsWRKY71* [69]. *WRKY30*, *WRKY4*, and *WRKY80* positively regulate SB resistance in rice [11]. In our study, 32 up-regulated WRKYs may act as positive regulators and 3 down-regulated WRKYs may act as negative regulators (Appendix A). Interestingly, WRKYs and AP2/EREBP TFs may synergistically regulate resistance to fungal diseases in rice. This view is supported by the research of Qiu et al. [69]. They found that AP2/EREBP proteins may be most involved in the control of *OsWRKY13* down-regulated genes, which is a negative transcriptional regulator [69].

The TIFY TF family is an important plant-specific gene family, and members of it were divided into JAZ, ZML, PPD, and TIFY subfamilies [70]. In our study, all 11 uDEGs in the TIFY family belong to the JAZ subfamily, consistent with the above results, and may be involved in the response to *R. solani* infection through JA signaling (Appendix A).

The plant mitochondrial transcription termination factor (mTERF) family belongs to the family of helical-repeat proteins. Although the function of plant mTERFs remains poorly elucidated, understanding the molecular mechanisms that regulate organellar gene expression is important [71]. The mTERFs have been implicated in the response to a variety of abiotic and biotic environmental stresses and in development by regulating the expression of organellar genes [71,72]. In our study, 11 down-regulated mTERFs responded to *R. solani* invasion in the resistant cultivar ZD; in addition, according to KEGG pathway enrichment analysis, dDEGs enriched chloroplasts. Thus, it is reasonable to assume that these mTERFs may act as negative regulators in disease resistance by regulating chloroplast gene expression. This may be an opportunity to study the function of mTERFs in biotic stress.

### 3.6. Chloroplast and Inhibition of Photosynthesis in Plant Immunity

Chloroplasts play a critical role in both plant productivity and the integration of the cellular response to stress and are therefore active sensors of the environment [73]. This is because they produce the largest pool of ROS of all cellular compartments, which are essential for the regulation of central redox homeostasis and retrograde signaling, and have been implicated in the up-regulation of defense-related genes and the down-regulation of photosynthesis genes [74,75]. Furthermore, chloroplasts also contribute to stress hormonal signaling by supplying biosynthetic precursors such as SA, JA, ABA, and ET [76]. In our study, the dDEGs were mainly concentrated in chloroplasts based on CC enrichment analysis. Our results support the idea that chloroplasts play an important role in disease resistance in plants, and also suggest that these dDEGs may have a significant effect on rice SB resistance as negative regulators.

When plants are infected, photosynthesis is inhibited as CO2 availability is reduced due to the closure of stomata, and in the apoplast, MAPK cascades are triggered, leading to the quick down-regulation of photosynthetic genes [74,77]. In our study, the dDEGs were enriched in the photosynthesis pathway when the resistant cultivar ZD was infected, but there was no response when the susceptible cultivar XZ was infected. Thus, the inhibition of photosynthesis is part of the resistance response.

## 4. Materials and Methods

### 4.1. Plant Materials and Growth Conditions

Twenty-two rice varieties screened for SB resistance seeds were obtained from the Tianjin Academy of Agricultural Sciences. Rice was grown under natural light in a net house in which the temperature ranged from 20 to 30 °C.

### 4.2. Pathogen Inoculation and Disease Scoring

The fungal strain of *R. solani* used in this assay, TJ-1, was provided by the Institute of Plant Protection of Tianjin Academy of Agricultural Sciences. Prior to inoculation, fungi were grown on Potato Sucrose Agar (PSA) for 3 to 4 days at 28 °C. The method of inoculation was carried out according to Yadav et al. [78]. Briefly, tillering stage plants were inoculated with *R. solani* by inserting the agar disk with mycelia slightly into the second sheath of rice. For each treatment, 10 plants were included. The temperature ranged from 30 °C to 35 °C and the humidity was >80%. The inoculated plants were evaluated for disease response as a percentage of the relative lesion height (RLH%) 4 weeks after inoculation, as shown below:$$\mathrm{RLH}\%=\frac{\mathrm{Lesion}\;\mathrm{height}\;\left(\mathrm{cm}\right)}{\mathrm{Plant}\;\mathrm{height}\;\left(\mathrm{cm}\right)}$$

Scoring was performed on the 0–9 scale of the Standard Evaluation System (SES) for rice (IRRI, 2002) [79].

### 4.3. Sample Collection, RNA Extraction, and RNA Sequencing

After inoculation for 48 h, obvious symptoms of SB were observed. Inoculated leaves of 22 rice varieties were cut from plants 48 h after inoculation, and uninoculated rice leaves were collected simultaneously as a control. Three plants per variety were pooled and immediately frozen in liquid nitrogen and stored at −80 °C. For each treatment, three independent replicates were included. The total RNA was extracted using Trizol reagent (Invitrogen, USA) according to the manufacturer’s instructions and then treated with DNase (Promega Corp., Madison, WI, USA). RNA concentration and quality were determined using the Agilent Bioanalyzer 2100 system (Agilent Technologies, Santa Clara, CA, USA) and a NanoDrop 2000 spectrophotometer (Thermo Fisher Scientific, Waltham, MA, USA), respectively. The sequencing of the transcriptome and the construction of cDNA libraries of all the samples were performed by GeneDenovo Biotechnology Co. (Guangzhou, China).

### 4.4. Differentially Expressed Genes Analysis and qPCR Validation

Clean reads from each RNA-seq sample were mapped to the rice reference genome IRGSP1.0 using the package HISAT2 (v2.1.0). The annotation data of genes were from RAP-db (https://rapdb.dna.affrc.go.jp/, accessed on 11 November 2021). The expected number of fragments per kilobase of transcript sequence per million base pairs sequenced (FPKM) values of each unigene were calculated using RSEM and further compared between groups using DESeq2 in the EdgeR package to represent relative expression levels. Differences with an absolute fold change of FPKM value ≥ 2, false discovery rate (FDR) < 0.05, and *p*-value < 0.05 were considered statistically significant; these genes were considered differentially expressed genes and further categorized into up-regulated and down-regulated expressed genes.

To validate the results of identified DEGs, qRT-PCR analysis was performed on 15 up-regulated and down-regulated genes. The rice gene *OsACTB* was used as an internal control. Specific primers were designed according to each gene sequence (Appendix A). cDNA was synthesized from total RNA (the same RNA samples for Illumina sequencing) using the Hifair Ⅱ 1st Strand cDNA Synthesis Kit (Yeasen Biotechnology, Shanghai, China). qRT-PCR analysis was performed using Hieff qPCR SYBR Green Master Mix (Yeasen Biotechnology, Shanghai, China) on an ABI Viia7 Real-time qPCR machine (Applied Biosystems, Foster City, CA, USA). Each reaction was repeated three times as technical replicates. Gene expression levels were calculated using the 2^−ΔΔ^CT method.

### 4.5. Functional Enrichment Analysis

The expression pattern of DEGs was normalized to 0, log2 (v1/v0), and log2 (v2/v0) and then clustered to different profiles using the Short Time-series Expression Miner (STEM) v0.75 software [80]. Enrichment analysis of up- and down-regulated DEGs was performed using the Web GO enrichment analysis tool ShinyGO (http://bioinformatics.sdstate.edu/go/, accessed on 26 November 2021), with enrichment FDR cut-offs of 0.05, based on GO terms of MF, CC, BP, and KEGG pathway.

### 4.6. Transcription Factors Analysis

Expressed genes and DEGs from RNAseq were used to analyze transcription factors according to their ID in RAP-db. Information on transcription factors in rice genomes with IDs of RAP-db and transcription factor families was obtained from the Rice TF Database (https://ricephylogenomics.ucdavis.edu/tf/, accessed on 27 July 2022) [81].

## 5. Conclusions

Transcriptome analysis was performed on resistant and susceptible rice varieties to investigate their response to *R. solani* infection. The results showed a significant difference in DEGs between the two types of varieties. Further analysis of uDEGs and dDEGs of the resistant varieties showed significant differences in gene category. The study found that genes related to PTI response, such as PRRs, the Ca ion signaling pathway, and the MAPK pathway, were up-regulated and played a positive role, and several genes related to JA and ET signaling pathways were activated and mainly positively regulated, while several genes related to the auxin pathway were down-regulated and mainly negatively regulated in response to *R. solani* invasion. In addition, many TFs are involved in the immune response to this pathogen. Furthermore, the study found that chloroplasts play a critical role in the response to *R. solani* invasion and that reduced photosynthetic capacity is an essential feature of the response. The results of this study are crucial for understanding the molecular mechanism of the response to *R. solani* invasion in disease-resistant rice varieties and can serve as an essential basis for the development of SB-resistant rice varieties.

## Figures and Tables

**Figure 1 ijms-24-14310-f001:**
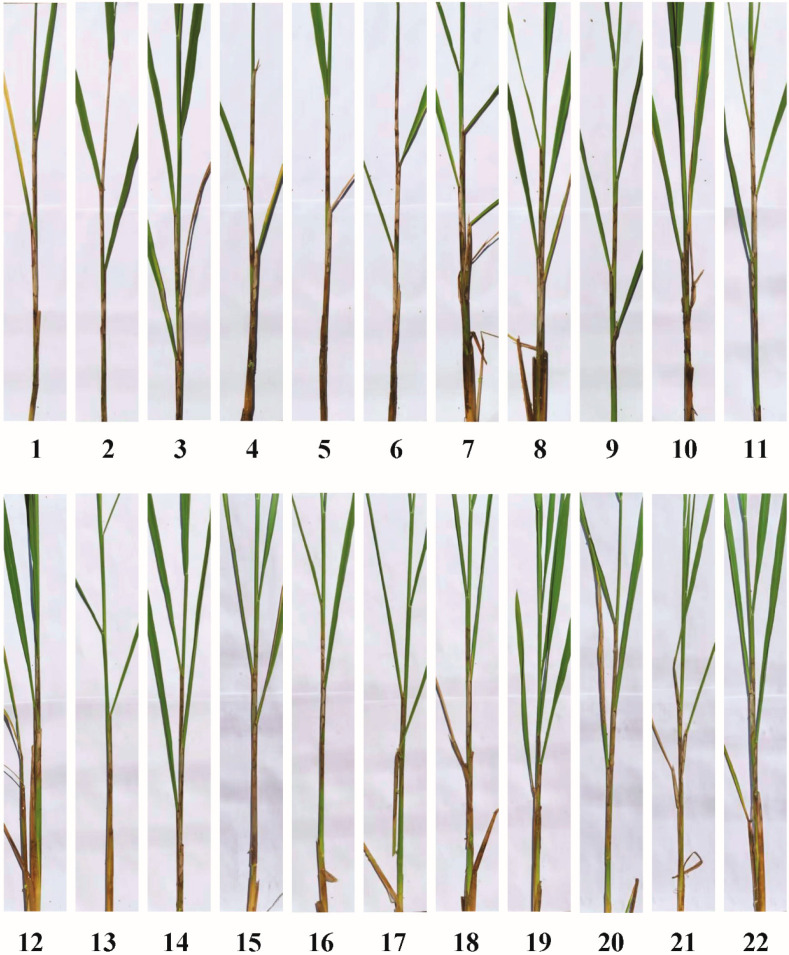
Symptoms of sheath blight of 22 rice cultivars after inoculation for 4 weeks. This figure is a supplementary material for Table 1; the numbers under each picture are described in Table 1.

**Figure 2 ijms-24-14310-f002:**
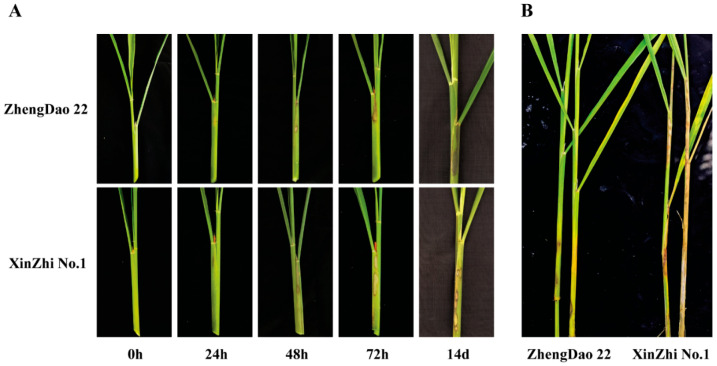
Phenotypes of resistant and susceptible cultivars against *R. solani* infection. (**A**) Phenotypes of resistant and susceptible cultivars against *R. solani* infection. (**B**) Phenotypes of resistant and susceptible cultivars under infected conditions at 10 weeks after inoculation.

**Figure 3 ijms-24-14310-f003:**
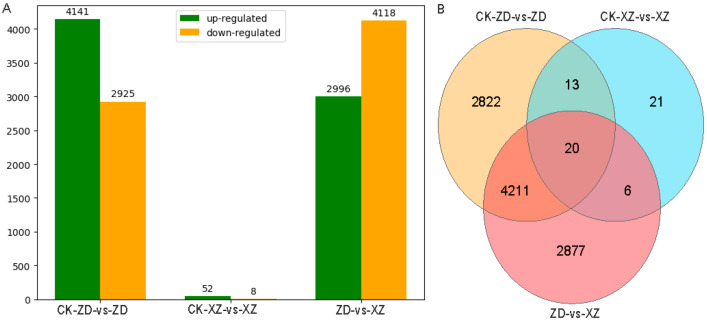
DEG response to *R. solani* in tolerant cultivar ZD and susceptible cultivar XZ. (**A**) The number of DEGs of different combinations. (**B**) Venn diagram comparing DEGs of different combinations. CK-ZD-vs-ZD is a comparison between uninfected and infected samples in the tolerant cultivar ZD. CK-XZ-vs-XZ is a comparison between uninfected and infected samples in the susceptible cultivar XZ. ZD-vs-XZ is a comparison between infected ZD and XZ.

**Figure 4 ijms-24-14310-f004:**
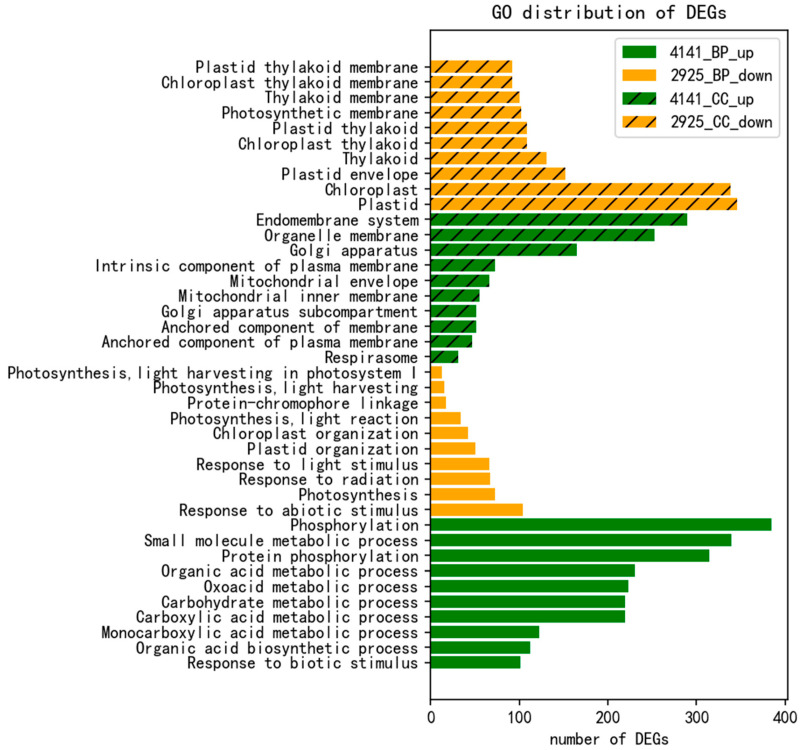
GO enrichment analysis of DEG response to *R. solani* in the tolerant cultivar ZD: 4141_BP_up is enrichment analysis of uDEGs based on BP of GO; 2925_BP_down is enrichment analysis of dDEGs based on BP of GO; 4141_CC_up is enrichment analysis of uDEGs based on CC of GO; 2925_CC_down is enrichment analysis of dDEGs based on CC of GO.

**Figure 5 ijms-24-14310-f005:**
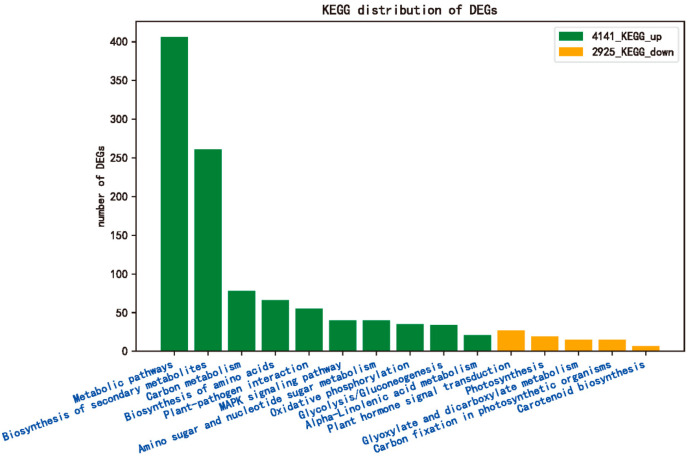
KEGG enrichment analysis of DEG response to *R. solani* in the tolerant cultivar ZD: 4141_KEGG_up is enrichment analysis of uDEGs based on KEGG; 2925_KEGG_down is enrichment analysis of dDEGs based on KEGG.

**Figure 6 ijms-24-14310-f006:**
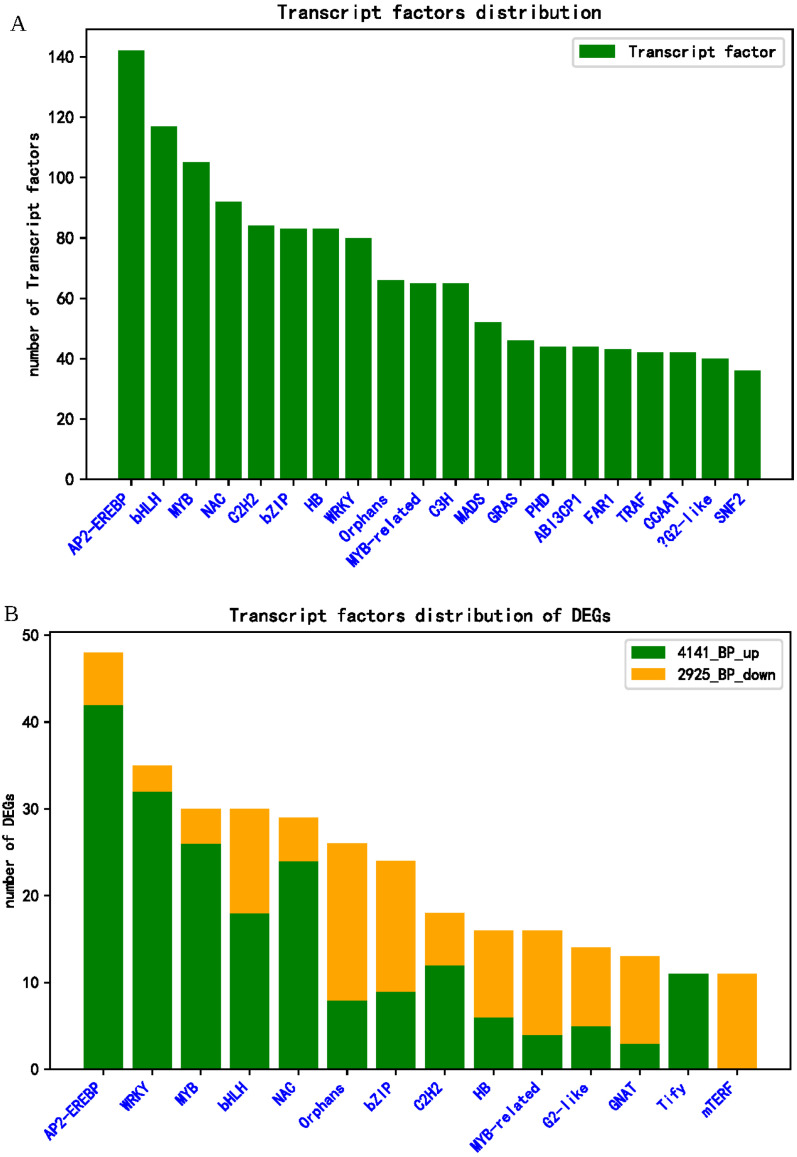
TF and DEGs of TF distribution. (**A**) Distribution of expressed transcription factors in RNA-seq. (**B**) Distribution of differentially expressed transcription factors: 4141_BP_up is uDEGs, 2925_BP_down is dDEGs. The x-axis represents the transcription factor family.

**Figure 7 ijms-24-14310-f007:**
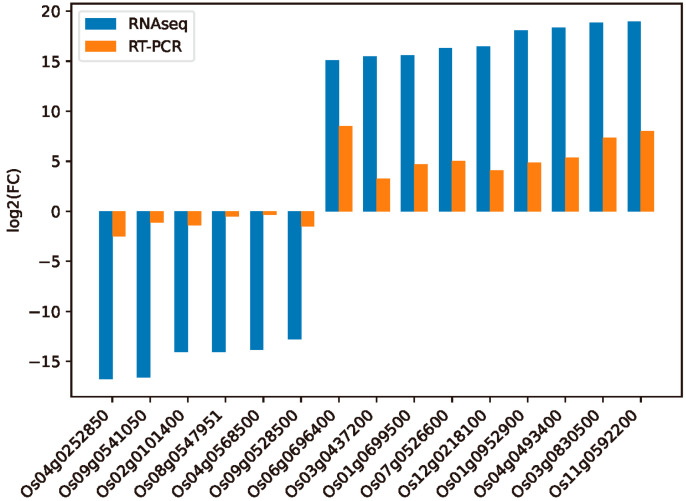
RT-PCR validation of differential gene expression results of RNA-seq.

**Table 1 ijms-24-14310-t001:** The resistance to SB in 22 rice cultivars according to relative lesion height.

No. of Sample	Cultivar	Relative Lesion Height (%)	Reaction *
1	JingHua 1832	56.0	S
2	XinZhi No.1	82.9	HS
3	XinDao 575	44.1	MS
4	XinFeng 21	50.0	S
5	XinLiang 501	52.4	S
6	ZhengDao 201	59.5	S
7	YiDao 178	67.2	HS
8	XinKeDao 37	52.1	S
9	YueNongDao No. 1	38.2	MS
10	ShengDao 735	34.3	MS
11	Xu72985	69.4	HS
12	YuanDao26	52.9	S
13	ZhengDao 22	14.5	MR
14	YuDao 24	63.2	S
15	XinKeDao 42	26.5	MR
16	LianDao819	48.8	S
17	XinLiang 320	29.8	MR
18	YiDao 675	38.2	MS
19	JingGuang17	47.4	S
20	LQ202	55.3	S
21	ZhengDao25	39.4	MS
22	XinDao No.18	51.4	S

* S means susceptible, HS means highly susceptible, MS means medium susceptible, MR means medium resistant.

**Table 2 ijms-24-14310-t002:** Statistics of differentially expressed genes.

Combination	Up-Regulated	Down-Regulated	All DEGs
CK-ZD-vs-ZD	4141	2925	7066
CK-XZ-vs-XZ	52	8	60
ZD-vs-XZ	2996	4118	7114

## Data Availability

The data presented in this study are available in the article and the Appendix A. The raw RNA-seq sequence data generated in this study have been submitted to the Genome Sequence Archive in China National Center for Bioinformation (https://ngdc.cncb.ac.cn/, accessed on 30 July 2023) under accession number CRA011415.

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
