# Peer review of "Comparison of Transcriptome between Tolerant and Susceptible Rice Cultivar Reveals Positive and Negative Regulators of Response to Rhizoctonia solani in Rice"

_ijms, 2023, doi:10.3390/ijms241814310_

Round 1

Reviewer 1 Report

Dear authors,

the paper you have presented on an agricultural crop - rice that is important to

the nutrition of almost half of he human population provides insight for enhancing production procedures in fighting against diseseas.

the description of sensitive and non-sensitive varieties provide contribution to existing findings. 

the presentation of results has solid scientific sounness and significance

therefore the paper needs only some technical adjustments: e.i indiscussion written in bold letters and set at the left margin

for all the figures introduce their names at the bottom (e.i . 6; 6A) of each figure and use palatino linotype font in all figures

kind regards

the reviewer

Reviewer 2 Report

General comments

The paper is quite interesting, however there are some parts that need to be improved, especially the Materials and methods part.

Specific comments

Materials and methods

This section should be further developed, especially sections 4.4 to 4.7.

Results

Tables should be self-explanatory.

Table 1. It should be stated what is meant by: S, HS, MS, MR, ...

When referring to Tables and Figures in some places it is written in bold (Line 134) and in other places without bold (Line 126). This should be standardised.

Discussion

The paragraph from Line 260 to Line 278 is written in bold. Why is that?

A space must be left after a full stop. For example Line 464 or Line 467. The wording should be checked throughout the text.

Reviewer 3 Report

In their article, the Authors report the results of the transcriptome analysis of different rice cultivars that differ in tolerance/susceptibility to infection by Rhizoctonia solani. 

Their study is very interesting, with possible innumerable repercussions given the importance and diffusion of this cultivation.

the aim of their study is well defined and organized, the experimental part is extensive and well described, the bibliography is updated and extensive. The conclusions are adequate to the results obtained. 

The discussion is very extensive and even divided into paragraphs, sometimes too didactic, almost as if it were a review. I allow myself to suggest to the authors for their future works of this kind to combine results and discussion, to better immediately appreciate results and their biological significance. 

Some notes for the Authors:

- Please number the references

- Please pay attention to spaces between word in the text and to to capital letters (i.e., golgi)

- Please define all abbreviations to the first citation (i.e., GO in the Abstract)

- In my opinion it would be better to move table 1 before figure 1 to understand which are the 22 analyzed cultivars. Furthermore, given that the materials and methods section is after the discussion, I suggest expanding the captions of table 1 (acronyms indicative of the reaction) and of figure 2. Or indicate something more in the text of the experimental section.

- I suggest rewording lines 48-49 and "of the phenylpropanoid pathways JA" at line 71 to make them more understandable

Minor editing of English language is required

Round 2

Reviewer 2 Report

The proposed changes have been made.